# communications
# engineering

# Superheated steam production from a large-scale latent heat storage system within a cogeneration plant

Maike Johnson [1][✉] & Michael Fiss[1]

During phase change, phase change materials absorb or release latent heat at a nearly constant temperature. Latent heat thus can be stored and integrated with evaporation/condensation systems such as steam generators within a relatively narrow range of operating temperature. Storage units and systems have been proven at pilot scale but none to-date have been integrated in industrial processes. This remains a challenge, due to the size of the systems and to hurdles in design, permission and build. Here we integrate a megawatt-scale latent heat storage into a cogeneration power plant in Wellesweiler-Neunkirchen, Saarland, Germany. The storage produced superheated steam for at least 15 min at more than 300 °C at a mass flow rate of 8 tonnes per hour. This provided thermal power at 5.46 MW and results in 1.9 MWh thermal capacity. Our study demonstrates the feasibility of using latent heat storage in the industrial production of superheated steam.

[1] DLR, Institute of Engineering Thermodynamics, Pfaffenwaldring 38-40, 70569 Stuttgart, Germany. [✉]email: maike.johnson@dlr.de

Thermal energy is used for residential purposes, but also for processing steam and other production needs in industrial processes. Thermal energy storage can be used in industrial processes and power plant systems to increase system flexibility, allowing for a time shift between energy demand and availability[1]. To this end, various types of thermal energy storage have been developed, from thermo-chemical systems to molten salt, solid matter, or latent heat, as discussed in depth by Steinmann[2].

In latent-heat storages, the storage material changes phase from solid to liquid during the charging or energy absorption phase of operation, and from liquid to solid during discharging, or energy release. During phase change, the phase change enthalpy or latent heat is absorbed or released at a nearly constant temperature. Due to this, latent-heat storage integrated into evaporation/condensation systems such as steam generators provide or store relatively large amounts of energy per unit volume within the narrow temperature range of evaporation/condensation. The systems can be cooled to ambient temperatures, as solidification is integral to the system, as opposed to in molten salt thermal energy storage systems. Thermo-chemical storage concepts are in an earlier stage of development; currently, material development for the movability of reactants is still an issue[3].

To date, latent-heat storages tested with evaporation in the heat transfer fluid (HTF) at up to 700 kW$_{th}$ power have been tested and published, among others, by Laing et al.[4], Garcia et al.[5], and Weller et al.[6], but none of these systems produces superheated steam, and were not integrated into operating industrial processes. In Garcia et al.[5], an extensive listing and discussion of the existing high-temperature latent-heat storages are well documented; this listing shows developmental steps in the technology but no integration outside of the laboratory environment.

In this article, the commissioning of a latent-heat thermal energy storage system for the production of superheated steam in an industrial setting is discussed. This was developed, built, and integrated into a cogeneration power plant in Welleswweiler-Neunkirchen, Saarland, Germany. The design of the system and the method including the simulation results are discussed in detail in ref. [7], and the build and integration in the plant in ref. [8].

## Methods

**System requirements.** The requirements of this specific integration are for standby operation, resulting in a needs-driven (as opposed to scheduled) discharging at full load for at least 15 min, which is followed by a non-critical charging. While standby operation is not uncommon, these requirements are likely rare enough to not lead to the development of a specific decarbonization technology for this field. This type of storage system can, however, be introduced for waste heat integration or peak-shaving operation in, for example, autoclave processes[2]. This can be as a solo-unit or with a modular design, allowing for serial or parallel operation of multiple storage units within one system. Specific markets are being researched in parallel to the technology development[9,10]. Current price developments for both manufacturing, energy, and materials make cost estimations for such a new technology difficult and should be carried out by industrial process owners and storage system manufacturers.

The power plant in Wellesweiler-Neunkirchen is a cogeneration plant, producing steam for industrial customers for process and heating purposes, as well as electricity for the grid. The plant is heat-driven and the primary steam generator is a gas turbine (GT), with 5.2 MW$_{el}$ and 8.5 MW$_{th}$ nominal powers, connected to a downstream heat recovery steam generator (HRSG). The HRSG feeds steam mains, supplying steam to several industrial customers. The steam directly emitted from the HRSG has a temperature of 350 °C; water injection reduces the temperature to just above 300 °C. The GT burns mine damp from the mine-damp network in Saarland[11]. A basic schematic of the power plant, showing the integrated storage system, is depicted in Fig. 1.

The customers require constant specific parameters of steam with a minimum of 300 °C, 25 bar, and at least 8 t h$^{-1}$. When the GT and HRSG cannot supply this, the standby system must assume steam supply. This is often due to fluctuations in mine damp quality or quantity. The steam must be provided by the standby system within the time that the HRSG produces steam

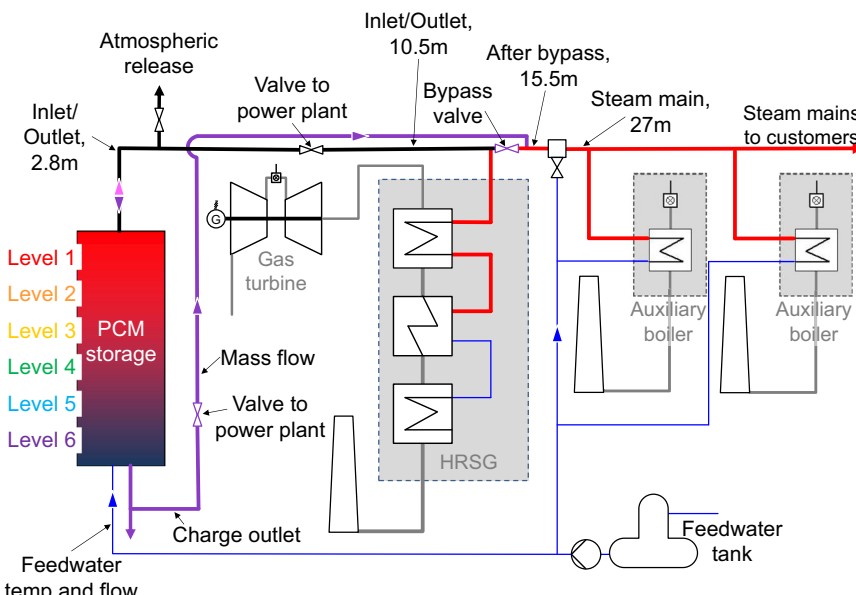

**Fig. 1 Simplified schematic of the integration of the storage.** The storage is in parallel to the heat recovery steam generator (HRSG) and auxiliary boilers with locations of the system and storage measurements denoted. Blue denotes feedwater and red denotes steam. Charging piping is shown in purple and discharging in pink.

after a GT-trip in lag-time, which is 2 min, as determined by experience from the power plant operator and discussed in ref. [7]. This ensures a constant steam quality in the steam mains.

Two auxiliary boilers, each with a nominal power rating of 13 $MW_{th}$, also feed the steam mains. The auxiliary boilers are used, depending on the season and related steam demand, either solely in standby or also to augment steam production. These auxiliary boilers can burn either mine damp or light fuel oil. In order to ensure high steam quality in the steam mains, an auxiliary boiler is held at least at minimal load. Minimal load is the lowest load at which fossil fuels are burned. Below this load, the component can be kept in so-called heat maintenance state; here, steam from another process (i.e., HRSG) flows through the auxiliary boiler to keep the component warm. From minimal load, full steam production can be attained in 2 min; from heat maintenance, 15 min are needed.

The thermal energy storage system is integrated into the power plant in order to reduce the minimal load operation of the auxiliary boilers. The fully charged storage can assume standby operation, which was to-date the operation in the minimal load of an auxiliary boiler. With the storage integrated, the auxiliary boilers are reduced from minimal load to heat maintenance.

The storage is, therefore, integrated in parallel to the HRSG and the auxiliary boilers (Fig. 1). When required, the storage is discharged and thereby evaporates feedwater to superheated steam. This steam flows past the water injection point to regulate the temperature to just over 300 °C and from there on to the steam mains. During this time, an auxiliary boiler is ramped-up from heat maintenance to full load.

The storage is discharged with 103 °C feedwater. The outlet parameter as required by the customers is steam at 300 °C, as stated. The saturation temperature at the system pressure of 25 bar is about 224 °C; the steam in the steam mains is, therefore, superheated by at least 76 °C. Using mass flow rates and enthalpy calculated with XSteam[12], which is based on IAPWS-IF97, the required minimum discharging parameters equate to a thermal power of 5.72 $MW_{th}$ using Eq. (1) and are summed to a thermal capacity of at least 1.43 $MWh_{th}$.

$$\dot{\mathbf{Q}}(t) = \dot{\mathbf{m}}(t) * \left(\mathbf{h}_{out}(t, p, T) - \mathbf{h}_{in}(t, p, T)\right) \quad (1)$$

Once the GT and HRSG are back in operation, the storage is charged, thereby returning it to standby conditions. During this time, an auxiliary boiler remains in minimal load operation, providing a standby for the HRSG/GT system until the storage system is again ready for operation. The storage is charged using steam directly from the HRSG. The steam can bypass the storage in total, which is normal operation, or in part (Fig. 1, purple), which is controlled by the bypass valve.

Steam flows downwards through the storage from the top to the bottom. At the beginning of charging, steam condenses. At the outlet of the lower header, the piping splits, with one part leading to a pressure release valve and a cooling pool for the condensate, and the other to the power plant system. Both of these piping sections are valve-controlled. The exiting steam temperature increases with the state-of-charge of the storage. Once the saturation temperature (~224 °C) is reached, the steam can be used by the power plant system; until this time, it is disposed of in the cooling pool. The mass flow rate going through the storage system is ramped-up during charging via a controlled bypass valve in order to maximize the steam used by the system. For most of the charging cycle, the steam cools in the storage but does not condense and is passed on to the customer. Because there is no phase change in the HTF during this time, comparatively little energy can be transferred, and charging, therefore, takes notably longer than discharging; the steam supply system continues operating while charging.

**Storage design**. In order to produce this superheated steam for these requirements, a thermal energy storage was developed. The development, detailed in ref. [7], resulted in a finned-tube shell-and-tube style latent-heat storage. The fins are assembled on the tubes with the clipping method studied by Johnson et al. in ref. [13]. Figure 2 shows the resulting hexagonal fin-tube assembly design. These fins are integrated to increase the heat transfer surface to the phase change material (PCM), which has a low thermal conductivity, especially when heat transfer is limited to conduction after solidification. The fin design results from a combination of two-dimensional simulations of heat transfer, empirical experience in manufacturing extruded aluminum, and experience with previous fin designs; the tube spacing in this storage is very dense at 70 mm.

Due to its melting temperature between the system limitations of 300 °C and 350 °C, as well as proven thermal stability as a PCM[14], sodium nitrate is used. The relevant material properties, as discussed by[14], are a theoretical melting temperature of 306 °C, a specific heat capacity of 1655 J $(kg\ K)^{-1}$, a heat conductivity of 0.55 W $(m\ K)^{-1}$, and a latent heat of 178 kJ $kg^{-1}$. The melting temperature of the actual salt inventory was measured using differential scanning calorimetry at about 304 °C (shown in the Supplementary Methods: Materials analysis, Supplementary

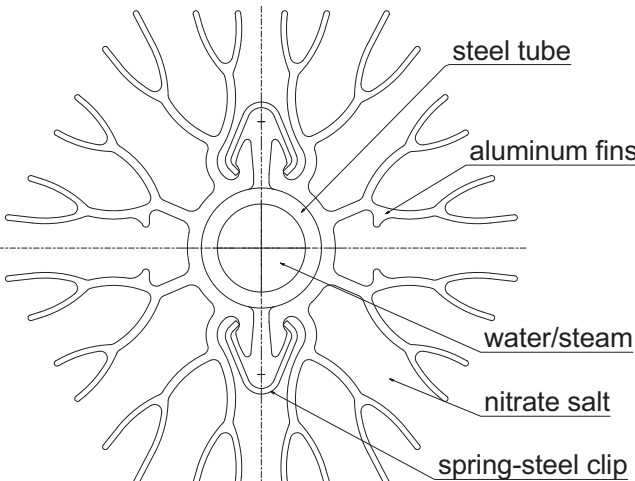

**Fig. 2 Diagram of fin-tube assembly.** Tree-like aluminum fins are held together by two spring-steel clips around a central steel tube.

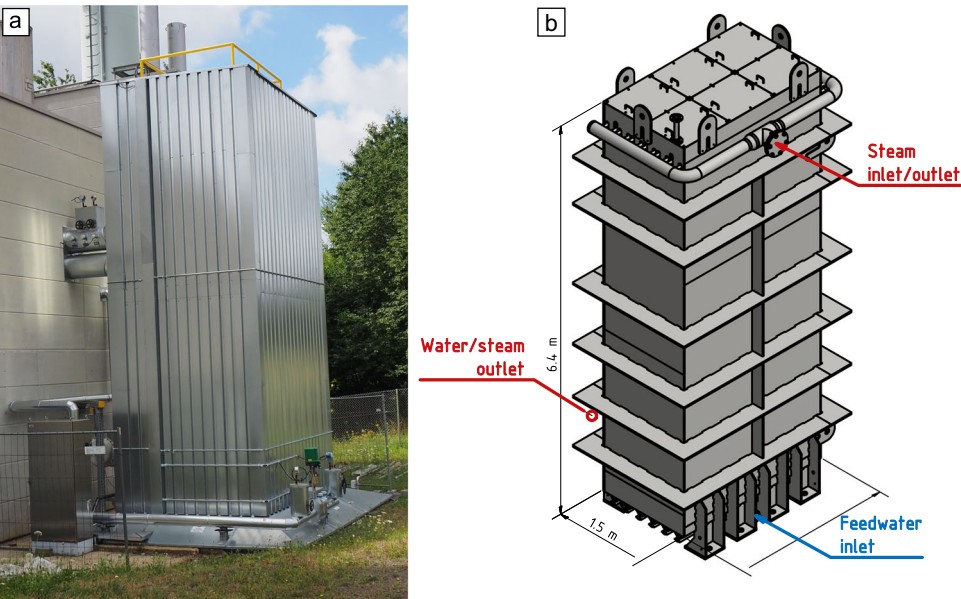

**Fig. 3 Cogeneration plant with integrated storage. a** A picture and **b** a schematic diagram. For reference: the control cabinet at the bottom left is 2 m tall.

| Table 1 Volume and fill level of the storage during iterative filling. | | | | |
|---|---|---|---|---|
| Filling iteration | Empty volume at 25 °C, in m³ | Salt mass at 350 °C in this volume, theoretical in t | Salt mass in storage, in t | Fill level, in % |
| Empty | 16.9 | 31.8 | 0 | 0 |
| After 1st | 8.9 | 13.8 | 18 | 57 |
| After 2nd | 4.7 | 4.2 | 27.6 | 87 |
| After 3rd | 3.5 | 1.5 | 30.3 | 95 |

Fig. 1). The ca. 32 t of storage material is on the shell side of the storage, directly contacting the fins. Figure 3 shows the storage installed, connected, and thermally insulated at the cogeneration plant, both as (a) a picture and (b) as a schematic diagram giving the basic dimensions.

**Storage filling**. Technical-grade sodium nitrate is the storage material in this system. This salt has a melting temperature of approximately 306 °C [14, and literature cited therein]. Due to the density differences in the salt depending on its temperature and form (Supplementary Methods: Materials analysis, Supplementary Table 2), filling is an iterative process. The iteration steps include manual filling, heating above the melting temperature, and active cooling as low as the system allows, followed by passive cooling down to safe working conditions.

After an initial filling of the storage, the system is heated above the melting temperature of the salt. For the heating step, the storage is heated from ambient temperature; during later normal operation, a return to ambient temperature is not planned. With the thermocouples dispersed in the salt volume of the storage, it is possible to measure the salt temperature at these points, but not globally throughout the storage. The storage was therefore heated with a low mass flow rate to slowly heat the system, and heated to well above the melting temperature. This is to ensure that all of the salt pellets melt. Due to the low thermal conductivity resulting from the point-form contact between the salt pellets, this process took several days for each iteration.

The heating of the storage was directly followed by an active cooling of the system. The active cooling is conducted by sending steam to the steam mains until the temperature is too low, and thereafter emitting the HTF to the atmosphere until the HTF

coming out of the roof expansion valve is in the liquid state. Thereafter, the system cools passively until safe working conditions are achieved; this passive cooling time increases with increasing fill level and takes 3–5 weeks.

Geometric calculations of the design of the storage results in a fill volume of 16.9 m³ of sodium nitrate, which results in a salt mass of approximately 32 t. However, considering the storage size, manufacturing tolerances and possible bowing of the side walls of the storage can change the ultimate fill level or amount. The iterative filling process has been conducted three times, and the storage attained approximately 95% fill level with the third filling, as detailed in Table 1.

**Data acquisition**. To analyze this storage system, both system data as well as detailed storage data are relevant. The system data measures the input-output of the storage system, including the flow parameters in different parts of the system but considering the storage itself more as a black-box. These results give insight into the performance of the whole system.

The storage measurements are used in order to better compare the design to the build of this storage. With this, it is possible to analyze solidification around the fins and at different points within the storage unit.

*System measurements*. The storage is integrated with two flanges at the bottom of the storage and one at the top, as shown schematically in Fig. 3b. At the bottom, one flange is an inlet for feedwater during discharging and the other is an outlet for water/steam during charging. The upper flange is a steam inlet/outlet for charging/discharging, respectively.

For the system analysis, the temperatures and pressures of the water/steam used as the HTF are measured near the flanges of the storage. The HTF temperatures are measured via 6 mm PT-100 type TR88, in 24 mm immersion sleeves type TW15, both from Endress and Hauser. The mass flow rate of the feedwater into the storage as well as the superheated steam in the steam main are measured. The feedwater flow is measured with an ultrasonic volumetric flow meter from Metra, type ultrakon®. In the steam main, the flow rate is measured via an orifice plate, and the temperature is measured via PT-100; these are components of an energy measurement unit from Metra, type autarkon® EWZ, coupled with a Danfoss IWK control unit. In addition, the ambient temperature is measured.

*Storage measurements.* For a detailed analysis of the storage, temperature measurements throughout the storage unit are used in combination with the system measurements. To this end, thermocouples are positioned throughout the PCM. In addition, thermocouples are affixed on the outside of the container walls, headers, and the outside of the insulation. The thermocouples are NiCr-Ni type K class 1 according to DIN EN 60584, with a 1.5 mm diameter Inconel sheath. Three lengths of thermocouple sheath were used in the system: 1.5, 6, and 9 m.

During the cycle analyzed here, 80 thermocouples were used for measurement. Of these thermocouples, 18 were calibrated according to DIN 55350-18-4.2.2 at the reference temperatures 103, 305, and 350 °C, correlating to the feedwater inlet temperature, the approximate melting temperature of the salt and the maximum steam inlet temperature, respectively. These calibrations showed an absolute minimum, maximum, and average deviation of respectively 0.01, 1.81, and 1.01 K.

The distribution of the thermocouples over the cross-section of the storage is shown in Fig. 4. Other thermocouples will be mounted during the finalization of the insulation. The thermocouples are affixed at six measurement levels. Some redundancy between the thermocouples exists, though not all measurements are fully redundant.

Thermocouples at the edges and corner of the storage—L02, B18, W18, M34, and W01—give insight into thermal losses over the large surface area of the storage and will assist in determining the necessity of recharging the storage if the standby time is long. In addition, there are thermocouples located between the edge and the center—L05 and S18—to also better understand thermal losses and temperature distribution throughout this large storage.

Most thermocouples are located in a central area around two central tubes. These are spaced so that the compromise between introducing an error by inserting a measurement device and needing the information is best met. The tube labels are those between columns I and N and rows 17 and 23, and are discussed in the Supplementary Discussion: Thermocouple results. A group of thermocouples near tube S5 is mounted specifically for an additional analysis of the state of charge, which will be conducted with further cycles.

The measurements are collected in a control cabinet next to the storage and are communicated from there to the power plant controls via a bus system. The control cabinet is on the north side of the storage, thereby having little temperature variation due to direct sun. The thermocouples are coupled into modules in the control cabinet of the multi-use type AI 731F from ABB with activated internal temperature compensation.

The data set and further analyses showed a residual current in the control cabinet that could not be adjusted for by the multi-use data acquisition modules used. These are not optimized specifically for thermocouples. As the measurement principle behind thermocouples is based on changes in voltage differences, the influence of the residual current has an influence on the data acquisition. To account for this, a length-dependent reductive shift was calculated into the thermocouple data. To determine the temperature-shift, the temperatures were measured with two thermocouples for each of the lengths with a calibrated hand-held measuring device and with the data acquisition system. The average difference between these two values was then subtracted from all thermocouple results according to their length.

The data for the specific storage system components is written for each data point every 30 s and transferred to files once a day. In the rest of the power plant, data is written to file when the value changes. The data analysis was conducted using Matlab® R2020a. Data for the thermocouple measurements is shown in the Supplementary Discussion: Thermocouple results.

## Results and discussion

**Charging.** These charging results were gained during commissioning, after the third filling of the storage with pelleted salt. It brings the storage system from ambient conditions to a charged state. The salt inventory is partially pelleted, meaning that a part of the inventory has only point-contact with other particles.

The charging data, shown in Fig. 5, show that charging was started at 9:33 on Nov. 26, and continued until 9:55 on Nov. 29, meaning the storage was charged for 72 h. Figure 5a shows the mass flow from the storage and the valve setting controlling flow to the storage. Figure 5b shows the temperature measurements at the storage inlet and outlet as well as after the bypass. Charging was started with an average temperature, measured by the thermocouples in the PCM, of 19.6 °C. The lowest measured temperature in the PCM was 17.3 °C, and the highest 22.3 °C. Steam was produced after about 3 h and after about 44 h, at 5:18 on Nov. 28, the outlet temperature was more than 300 °C, the required temperature for sending the steam directly to the steam mains without mixing. The mass flow rate during charging was about 4 t h$^{-1}$.

During charging, the storage is heated to well above 306 °C, the melting temperature of the sodium nitrate. Charging was completed with an average temperature measured by the thermocouples in the PCM of 327.9 °C. The highest measured PCM temperature was 331.5 °C and the lowest was 324.8 °C. This allows for a high driving temperature difference between the PCM and the feedwater during discharging.

**Discharging.** The storage is discharged using feedwater, which evaporates and is superheated, as shown schematically in Fig. 1 in blue and pink. The main system data results are shown in Fig. 6. Here, the feedwater mass flow (a, blue dashes) and the steam main mass flow rate (a, black line) are shown. The feedwater flow rate is an input to the system, and is a variable that is always varied by the operator according to power plant needs and operating conditions; for these initial tests, it is controlled manually. The five valves controlling flow to and from the storage in the different connections also are adjusted, leading to temperature reactions in the system, shown in Fig. 6b. With more operation experience and increasing automation of the discharge, the input flow rate will become more constant.

The steam temperature data is shown for several positions in Fig. 6b. Shown are the temperature measurements 2.8 m after the outlet of the storage, the temperature past the bypass, located about 15.5 m from the outlet of the storage and inside the GT-Building, and past the water injection site at the entrance to the steam mains, 27 m from the storage flange. These distances are given to assist in understanding differences in temperatures as well as shifts over time, considering the ambient temperatures (0.2 °C at the beginning of discharging).

For the discharging of the system after the third filling, discharging began at 10:11, at which time mass flow into the storage started and the valves to the storage and from the storage

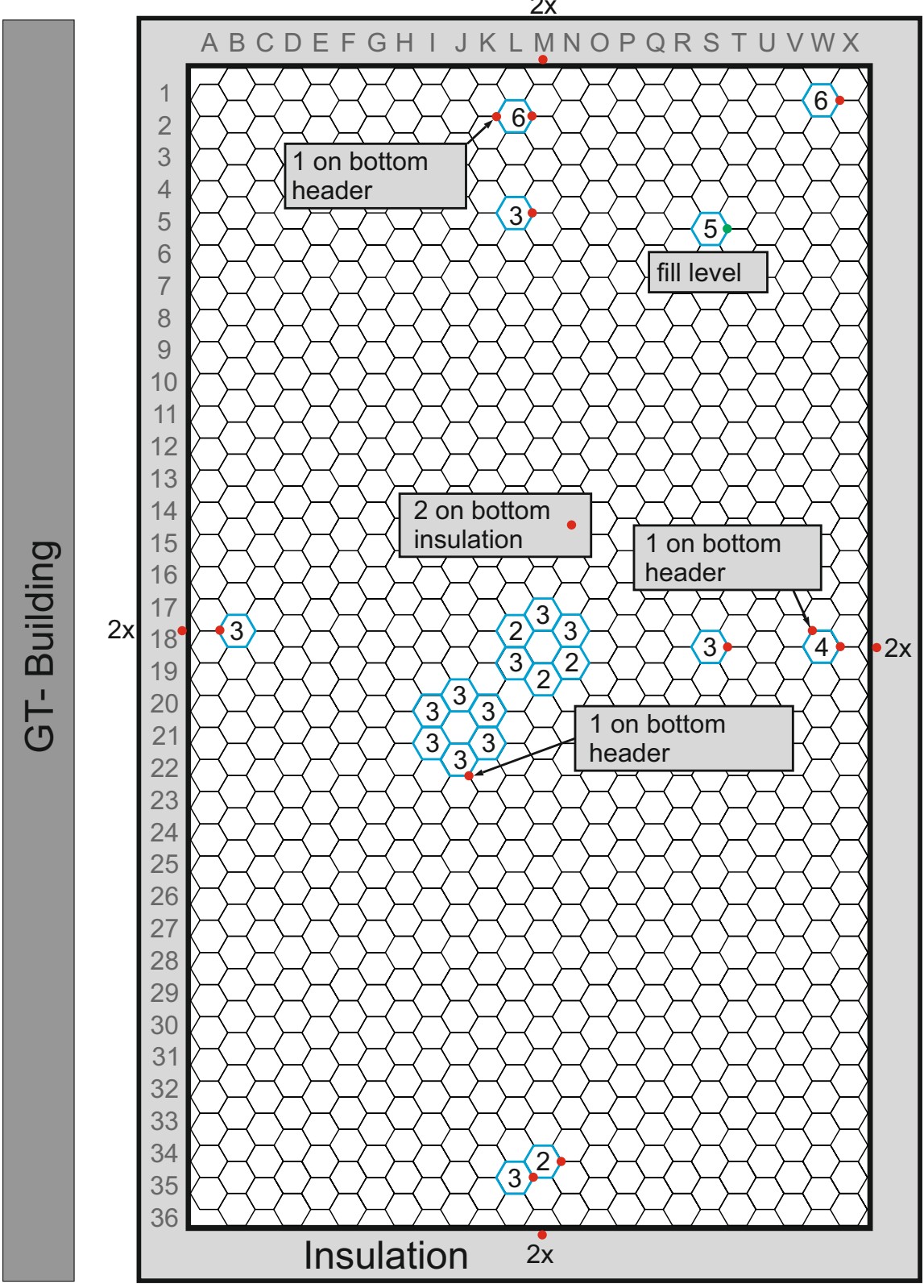

**Fig. 4 Schematic location of thermocouples.** Shown from above is a cross-section of the storage, with row numbers and column letters for location, and finned-tubes denoted by hexagons. Numbers in hexagons denote how many vertical levels are equipped with thermocouples. Thermocouples are mounted around two central tubes for redundancy, as well as at the middle, edges, and corners. Red dots denote thermocouple locations, also on the container wall at two heights.

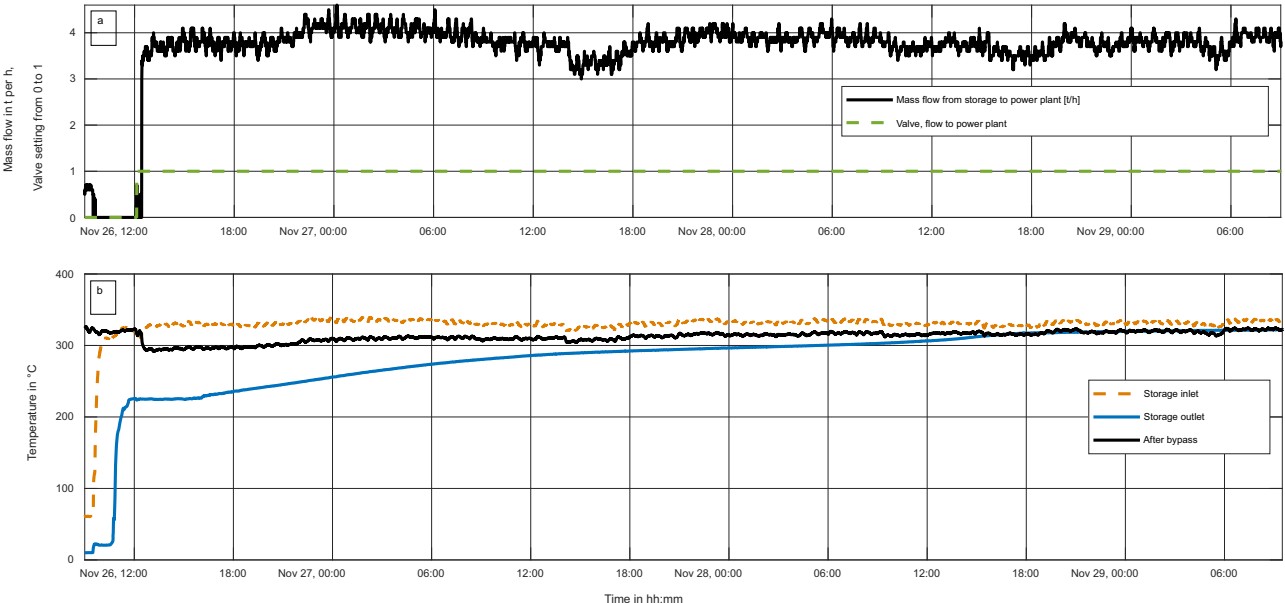

**Fig. 5 Plot of charging data of the storage system.** Shown from ambient conditions, showing **a** mass flow rate through the system and the valve setting denoting when the flow through the storage system flowed to the steam mains and was no longer emptied to the condensate outlet as well as **b** temperatures of the storage inlet and outlet near the top and bottom flanges, respectively, and after the system bypass.

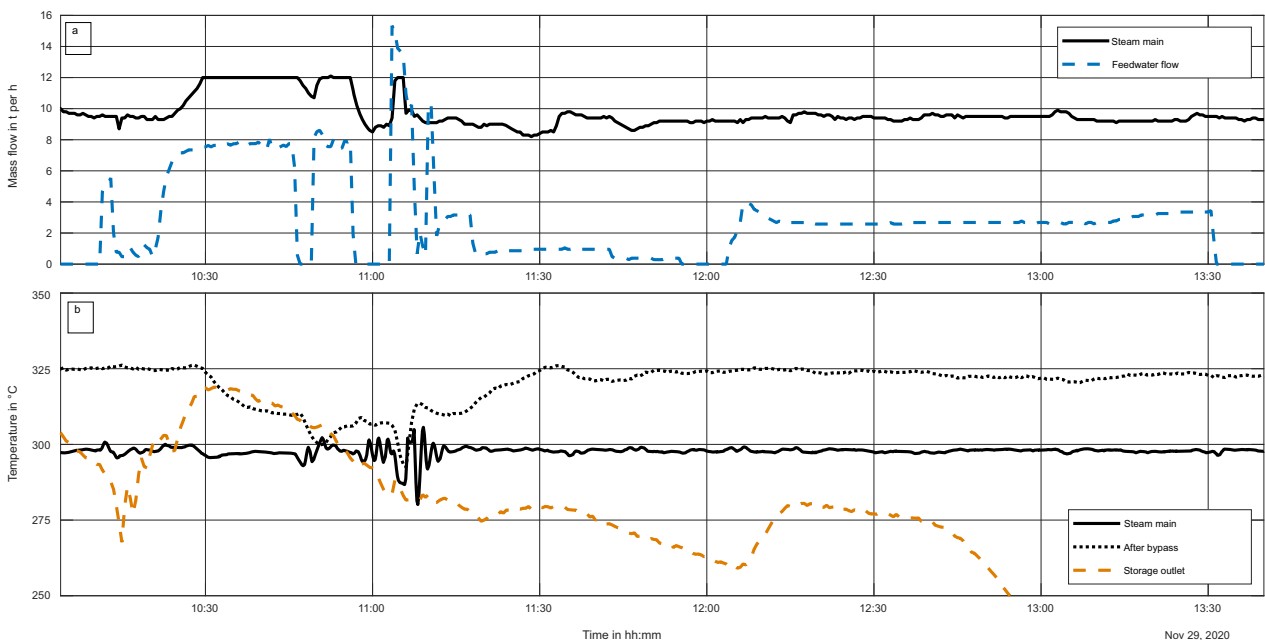

**Fig. 6 Plot of full discharging of the storage system.** Showing **a** mass flow rate through the storage system and in the steam main and **b** temperatures of the steam past the outlet of the storage, just after the system bypass and in the steam main.

to the power plant were opened. At 13:33, feedwater flow into the storage system was stopped.

The minimum required temperature for stable operation of the steam mains is 300 °C. In a real operation of the system, the HRSG would not be in operation, so this requirement must be met by the storage system. This temperature was maintained for 28 min (from 10:25 until 10:53). The first 20 min (from 10:25 until 10:45), the mass flow rate through the storage was approximately the nominal rate of 8 t h$^{-1}$; this means that during this timeframe, the storage was discharged under nearly nominal conditions.

The measurement position for the mass flow rate in the steam main measures the combined flow from the storage and the HRSG. During this initial operation, the HRSG was also in operation, so that the mass flow rate in the steam mains is augmented by the steam from the storage system; steam production by the HRSG was manually adjusted to match the demand parameters.

The steam from the storage system was fed into the main from 10:22 until 11:04. At 11:04, the steam temperature after the bypass sunk to 300 °C. At this point, the valve to the main was closed and the flow was emitted to the atmosphere. Discharging continued in

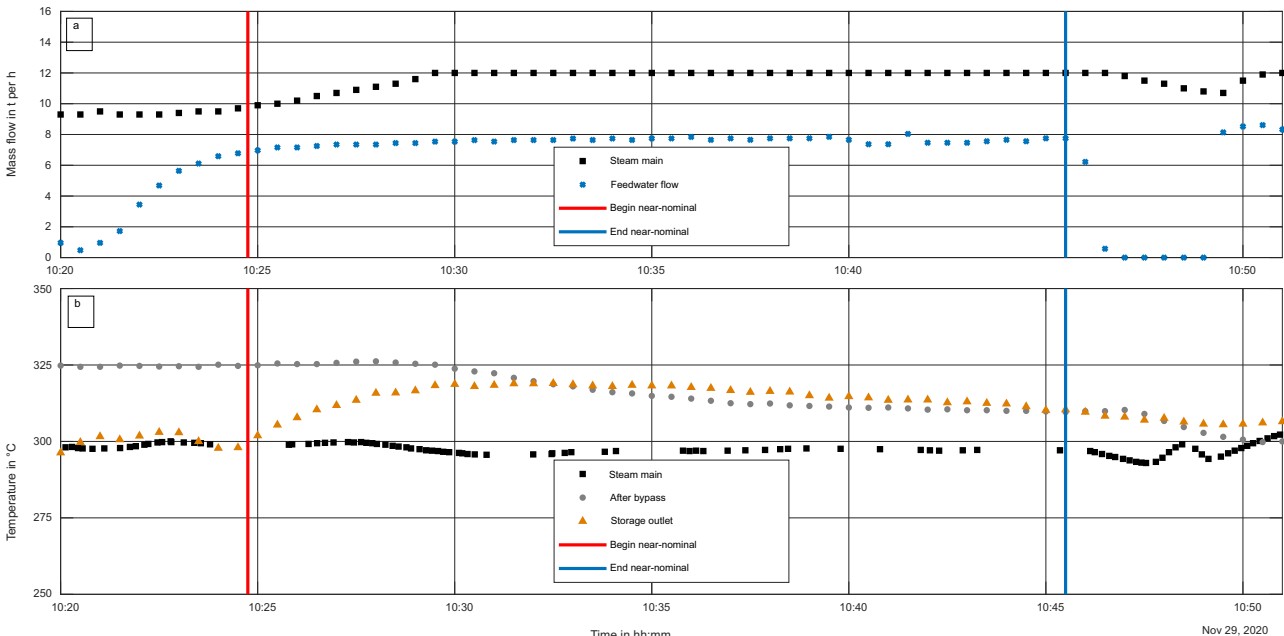

**Fig. 7 Plot of discharging of the storage system for near-nominal conditions.** Near-nominal conditions are denoted by vertical lines, from 10:20 to 10:50, showing **a** mass flow through the storage system and in the steam main and **b** temperatures of the steam past the outlet of the storage, just after the system bypass and in the steam main.

order to cool the storage as much as possible for further commissioning, as described in the methods section.

The system requirement for the storage system is the production of superheated steam for at least 15 min; during this discharging, steam of sufficient quality (>300 °C, >25 bar) was produced for 43 min, showing a much better performance than necessary.

**Thermal power and capacity**. The discharging of the storage system is the more critical of the two processes. It requires high thermal power from the component, and only with this thermal power can the system parameters be reached and the storage be useful for the power plant once the testing phase is completed. This high thermal power level requires a very fast heat transfer from the PCM to the HTF.

To assess the capabilities of the storage system, the results were analyzed for the time during which the outlet temperature of the storage was greater than 300 °C and the mass flow rate close to 8 t h$^{-1}$. During this timeframe, the average mass flow rate of feedwater into the storage was 7.6 t h$^{-1}$, with a minimum of 7.0 t h$^{-1}$ and a maximum of 8.0 t h$^{-1}$. The timeframe for these near-nominal conditions was 10:25 and 10:45. The details for this timeframe are shown in Fig. 7 in (a) for the mass flow rates and (b) for the temperatures.

From the measurements, the thermal power transferred to the HTF is calculated using Eq. (1). For each time step of 30 s, using the temperature and pressure data, enthalpy is calculated at the inlet and outlet of the storage using XSteam[12]. With this and the mass flow rate, thermal power is calculated and summed to thermal capacity.

These results are plotted in Fig. 8. The timeframe for capacity summation, shown in (c), begins with the temperature meeting the 300 °C criterion (orange vertical line, temperature shown in (b)) and ends when the power plant operator down-regulated the mass flow rate (blue vertical line, mass flow shown in (a)).

The average discharging power during this time was 5.46 MW$_{th}$; the energy discharged was 1.9 MWh$_{th}$. As discussed earlier, the operating requirements lead to a necessary power of 5.72 MW$_{th}$ and a capacity with a 15-min discharge of 1.43 MWh$_{th}$. This thermal power level was not quite attained; however, with a mass flow rate of 8 t h$^{-1}$ and not just 7.6 t h$^{-1}$, the requirements will be met.

## Conclusions

The storage system analyzed here is integrated into an operating cogeneration plant, mostly providing steam to critical customers. This gives insight into the entire process of design, permitting, build, and integration into a real system. However, the parameters with which the system can be commissioned and tested are limited by both power plant system limitations as well as operating restrictions, as the steam demand has to be met constantly.

Within this work, a storage system for the production of superheated steam for an industrial system was operated. This process is a cogeneration plant requiring steam with at least a temperature of 300 °C and a pressure of 25 bar. In order to maintain steam quality in the steam mains for the industrial processes, this storage is integrated as a standby for the GT and HRSG, providing the full steam load of 8 t h$^{-1}$ within 2 min if the GT trips, and providing this steam for at least 15 min while an auxiliary boiler increases load from heat maintenance to full load. This integration allows for a load reduction in the auxiliary boilers from minimal load, which burns fossil fuels, to heat maintenance, reducing the use of oil as a primary energy source.

The results from an initial operation are discussed. Charging of the storage took about 44 h from ambient conditions with pelleted salt. According to the design, from normal operating conditions post-commissioning, charging takes about 14 h. This is because charging is conducted using the sensible heat from steam, and that steam is used by the customer. Therefore, the system continues to operate during charging. In other integrations in which the condensate can be integrated into the industrial system, faster charging with phase change in the HTF would be possible.

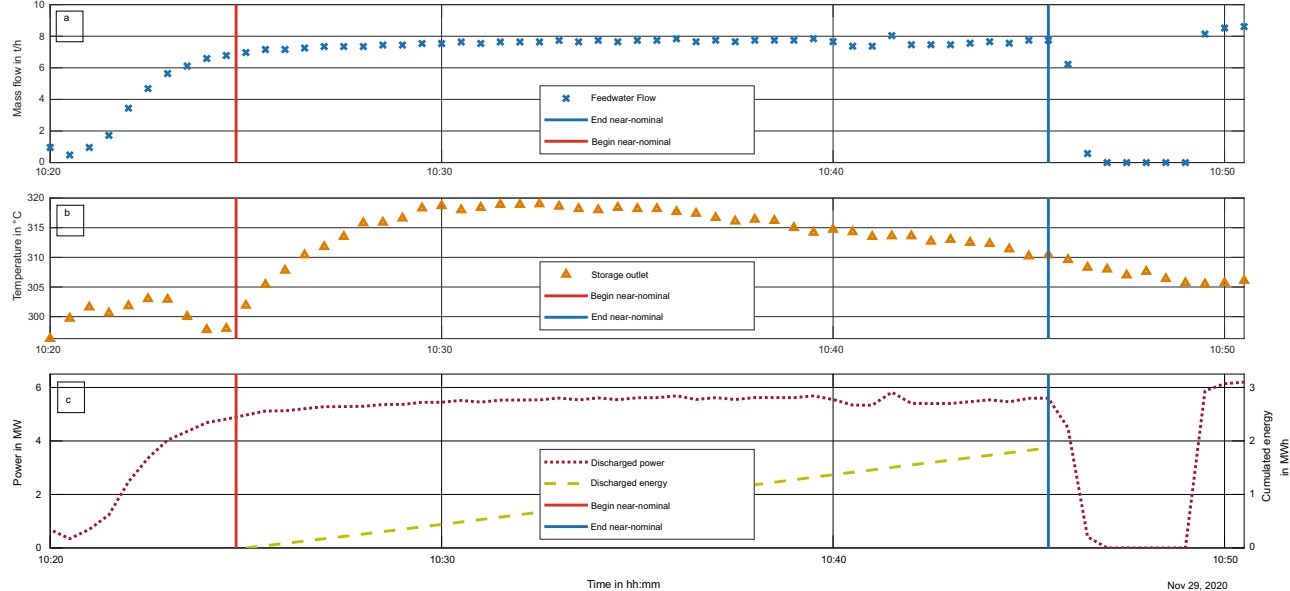

**Fig. 8 Plot of discharging of the storage system for near-nominal conditions from 10:20 to 10:50.** Near-nominal conditions denoted by vertical lines, limited by mass flow averaging 7.6 t h⁻¹ (shown in **a**) and an outlet temperature >300 °C (shown in **b**). These are calculated to power and energy (shown in **c**). Energy is cumulated during the near-nominal conditions.

The discharging process analyzed here directly followed a charging process. During commissioning, the method for recharging the storage is determined—either constantly recharged with a minimal flow or periodically. Both are possible during standby operations.

During this discharging, near-nominal conditions were attained for 20 min, with an average superheated steam mass flow rate of 7.6 t h⁻¹ to the industrial customers produced by the storage system. This exceeds the minimum requirement by 5 min, respectively, 33%. During this time, an average power of 5.46 MW$_{th}$ was discharged, resulting in an energy discharge of 1.9 MWh$_{th}$. The design goals were 5.72 MW$_{th}$ and 1.43 MWh$_{th}$, respectively; the flow rate used in this discharge was somewhat under the nominal flow rate of 8 t h⁻¹.

These results show that the high thermal power rates required in industrial processes are possible, overcoming the hurdle of heat transfer into PCMs. In addition, with more trust in the design methods used, smaller storages can be built to meet these same requirements; simply said, this storage is larger than necessary. However, as this was the first such storage system integrated into an operating system, safety factors were planned into the design.

Due to complications with the system integration, commissioning has been paused. Once commenced, it will embody a full testing regime of further charging and discharging cycles and tests at partial load, analysis of environmental losses, and more detailed comparisons of the thermocouple data with design results.

Although the integration goal of serving as a standby, and amortizing through the reduced use in fossil fuel use is less common, the need for peak shaving and other similar needs for steam delivery are more common. This first installation in an industrial setting can be used to reduce costs in future installations and better trust design methods for future storage systems specifically for different processes.

This storage system will not only provide system flexibility and fuel savings for this specific power plant system but, as the world's largest evaporative latent-heat storage system, it also shows that the feasibility of both upscaling, production of superheated steam and megawatt-scale heat transfer rates, as well as permitting a real system.

## Data availability

Data sets generated during the current study are available from the corresponding author upon reasonable request.

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

## Acknowledgements
The authors would like to thank the German Federal Ministry of Economic Affairs and Energy for partially funding the work within the TESIN project (Contract No. 03ESP011A). The authors are solely responsible for the content of this publication. The authors would like to thank Andrea Hanke from DLR for the differential scanning calorimetry measurements and Richard Meiers from ABB and Christoph Haupenthal from Iqony Energies for their support in understanding the residual currents in the control box. In addition, the authors would like to thank the Iqony Energies project colleagues for their continued assistance in transmitting data sets and answering research questions.

## Author contributions
M.J. is responsible for the conceptualization, data analysis, funding acquisition, methodology, project administration, validation, visualization, and writing of this paper. M.F. assisted in the formal analysis and discussion of methodology.

## Competing interests
The authors declare no competing interests.
