## [Peer Review File · Communications Engineering]

Reviewers' comments:

Reviewer #1 (Remarks to the Author):

The following issues should be addressed:

- 1- The thermal resilience of the thermal storage PCM should be studied. I suggest to include more explanation based on the DSC tests to comment on the life cycle of the material.
- 2- The CFD simulation of the finned storage requires experimental validation.
- 3- The repeatability of the experimental data should be reported.
- 4- The technical specifications of the integrated power generation system should be added.
- 5- The water mass flow rate variation should be illustrated in a figure in accordance with the power production profile in order to justify the time dependent variations.

Reviewer #2 (Remarks to the Author):

General Comments:

The paper subject matter is interesting and relevant to the field of thermal energy storage. It should be published eventually as demonstration data is important to disseminate, however, it does need significant revisions for a Nature journal. In general, de-risking technology by deploying larger-scale test beds is highly commendable and much needed. In addition to the technical content of the paper needing revisions, the importance and the case for thermal energy storage of this kind (capacity level) is not well made. Yes, TES is needed for many applications, but I frequently found myself wondering if 15-43 minute discharge capacities were valuable to various industrial applications. No discussion of what industry application heat quality and magnitudes were ever given. General statements are made by the authors regarding the motivation for TES, but not specific ones as it pertains to the needed TES energy storage capacity and durations for numerous applications such as breweries, steel-making, paper-making, etc. What do industrial facilities need? How much is it worth? What are the cost requirements (\$/kWh-thermal) that make TES attractive. Are there minimum charging requirements that industrial facilities need. The system here takes very long to charge. I would also guess the capacity needs are much, much larger than the 5 MWh demonstrated here, by at least a factor of 10-50. Thus, more discussion and context for the importance of this work is needed because it is hard to see how the thinking around TES and demonstrations are significantly influenced by what is reported and claimed herein.

With regard to the technical content, much of the paper would benefit from a re-write. The paper title is not meaningful and the manuscript reads more like project report, an article in a trade magazine, or a first draft conference paper. The results are also very specific, where discussion for example spends much time with details of pipe lengths, etc. Instead, the authors should generalize these descriptions

and provide such details in the Supplemental information section. For example, there is a detail within the manuscript (p.5) that the steam main temperature needs to be 300C and the measurement for this condition must be taken "2.8m past the flange." However, if such detail is to be given, discussion or explanation as to why 2.8m downstream should be given. Are there thermal (e.g., heat loss considerations in W/m of pipe length), or fluid dynamic (e.g., L/D minimums for stable flow measurements) considerations that govern this? Another example is that the ambient temperature of 0.2C is reported suddenly in the last paragraph above Fig. 5, but why it is reported is unclear. This happens frequently throughout the text and the readers would benefit from general guidelines and learnings of operating thermal plants than the several specifics offered without supporting explanations as to their particular significance.

Further, there are a low number of citations, and the paper manuscript organization is odd. Admittedly Nature CommEng requires the authors to place the Methods section after the Results and Discussion, but there are few clear section titles that follow the journal guidelines and are sorely needed, namely: Introduction, Results, Discussion, Methods, and Conclusions. The authors also need to review their section/subsection titles and refine the organization more clearly.

The Conclusions section makes some statements regarding the importance of "this data" to the research community – but which data exactly is important and why is not given? What information is needed to de-risk the technology? What specific techno-economic barriers do industrial users need to be overcome have before adopting TES? That is, claims are made to the importance of this work and data presented but not really supported. For example, what does the readership now know that it critically did not know before reading this work as it pertains to TES and large-scale demonstrations?

Detailed Comments:

1. The Title is not meaningful. What about "superheated steam discharged" from MW TES? It reads like an unfinished phrase or point yet to be made. What about "Demonstration of superheated steam generation from MW-scale latent TES using molten salts"? or some such variation.
2. Abstract – the authors refer to "power" levels – electrical or thermal? Be specific.
3. Abstract – the last sentences needs ", respectively" to be added to it.
4. Main, p. 2 - 3rd paragraph last sentence – what is the point of the last sentence? It is not clear why the readers should want to know that Garcia et al, have an extensive listing and discussion of latent heat storage at this particular location within your manuscript. Rather, a point should be made that is supported by Garcia, et al., and the reader can examine that reference more carefully is they want details on x, y or z.
5. Main - How is thermal energy rate ("power") defined? Provide definition.
6. Main, p. 2 – there is no reference for the two minute specification on GT lag time. Please provide.
7. Text around Fig. 2 - The fin-tube assembly geometry is very intriguing. Recommend including 1-2

sentences on how this was developed (ie the geometry was established by what kind of objective function and conducting heat in/out of TES is a critically important challenge).

8. p. 3 – sodium nitrate for the heat transfer fluid is given, but its thermophysical properties, namely is enthalpy change upon solidification and melting are not provided. Was this not determined from DSC? The supplemental information section provides no information on such thermodynamic property data obtained.

9. p.3 – Why was sodium nitrate chosen as the HTF?

10. Power capacity – “Thermal” power is a more accurate term. 2nd paragraph, last sentence – believe there is a type – “mass low” should be “mass flow”

Reviewers' comments: Reviewer #1 (Remarks to the Author):

Thank you for your time and insight. The authors hope the amendments and changes based on the reviewer's comments help clarify the article.

The following issues should be addressed:

- 1- The thermal resilience of the thermal storage PCM should be studied. I suggest to include more explanation based on the DSC tests to comment on the life cycle of the material.

A clearer reference to materials research on sodium nitrate was included in the section regarding system design and integration, as well as to the DSC measurements in the supplementary information. The section now reads:

Due to its melting temperature between the system limitations of 300 °C and 350 °C, as well as proven thermal stability as a phase change material (PCM) [11], sodium nitrate is used. The relevant material properties, as discussed by [11], are a theoretical melting temperature of 306 °C, a specific heat capacity of 1655 J/(kg K), a heat conductivity of 0.55 W/(m K) and a latent heat of 178 kJ/kg. The melting temperature of the actual salt inventory was measured using differential scanning calorimetry as about 304 °C (supplementary information, Fig. 9).

- 2- The CFD simulation of the finned storage requires experimental validation.

The CFD simulation cannot be validated in a detailed manner in a storage of these dimensions. A qualitative comparison is given in the supplementary information. Further validation research is currently being conducted, strengthening the confidence in the modeling method.

- 3- The repeatability of the experimental data should be reported.

A note discussing the lack of repeatability data was added to the conclusions and reads as follows:

Due to complications with the system integration, commissioning has been paused. Once commenced, it will embody a full testing regime of further charging and discharging cycles and tests at partial load, analysis of environmental losses and more detailed comparisons of the thermocouple data with design results. Currently, only this data set exists due to complications. The authors consider this one data set important as is, and wish to share the data with the community, even though further data sets are currently not available.

- 4- The technical specifications of the integrated power generation system should be added.

Technical specifications for the gas turbine and auxiliary boilers were added in the section System design and integration. The information is dispersed in the section and therefore not copied into this document.

- 5- The water mass flow rate variation should be illustrated in a figure in accordance with the power production profile in order to justify the time dependent variations.

A new Fig. 6 was included with a detail of the relevant time frame discussed in the power and capacity diagram, now Fig. 7.

Reviewers' comments: Reviewer #2 (Remarks to the Author):

Thank you for your time and insight. The authors hope the amendments and changes based on the reviewer's comments help clarify the article.

General Comments:

The paper subject matter is interesting and relevant to the field of thermal energy storage. It should be published eventually as demonstration data is important to disseminate, however, it does need significant revisions for a Nature journal. In general, de-risking technology by deploying larger-scale test beds is highly commendable and much needed. In addition to the technical content of the paper needing revisions, the importance and the case for thermal energy storage of this kind (capacity level) is not well made. Yes, TES is needed for many applications, but I frequently found myself wondering if 15-43 minute discharge capacities were valuable to various industrial applications. No discussion of what industry application heat quality and magnitudes were ever given. General statements are made by the authors regarding the motivation for TES, but not specific ones as it pertains to the needed TES energy storage capacity and durations for numerous applications such as breweries, steel-making, paper-making, etc. What do industrial facilities need? How much is it worth? What are the cost requirements (\$/kWh-thermal) that make TES attractive. Are there minimum charging requirements that industrial facilities need. The system here takes very long to charge. I would also guess the capacity needs are much, much larger than the 5 MWh demonstrated here, by at least a factor of 10-50. Thus, more discussion and context for the importance of this work is needed because it is hard to see how the thinking around TES and demonstrations are significantly influenced by what is reported and claimed herein.

Thank you for your interest! Further explanations and clarifications to the need for the technology are included in the introduction as a motivation, in addition to the explanations already in the conclusions.

With regard to the technical content, much of the paper would benefit from a re-write. The paper title is not meaningful and the manuscript reads more like project report, an article in a trade magazine, or a first draft conference paper. The results are also very specific, where discussion for example spends much time with details of pipe lengths, etc. Instead, the authors should generalize these descriptions and provide such details in the Supplemental information section. For example, there is a detail within the manuscript (p.5) that the steam main temperature needs to be 300C and the measurement for this condition must be taken "2.8m past the flange." However, if such detail is to be given, discussion or explanation as to why 2.8m downstream should be given. Are there thermal (e.g., heat loss considerations in W/m of pipe length), or fluid dynamic (e.g., L/D minimums for stable flow measurements) considerations that govern this? Another example is that the ambient temperature of 0.2C is reported suddenly in the last paragraph above Fig. 5, but why it is reported is unclear. This happens frequently throughout the text and the readers would benefit from general guidelines and learnings of operating thermal plants than the several specifics offered without supporting explanations as to their particular significance.

The authors reviewed and edited the article, removing some of the details and explaining the relevance of other details. As these are dispersed throughout the article, the changes are not quoted here.

Further, there are a low number of citations, and the paper manuscript organization is odd. Admittedly Nature CommEng requires the authors to place the Methods section after the Results and Discussion, but there are few clear section titles that follow the journal guidelines and are sorely

needed, namely: Introduction, Results, Discussion, Methods, and Conclusions. The authors also need to review their section/subsection titles and refine the organization more clearly.

Thank you for this insight. The heading ,Results‘ somehow got misplaced and has been reinserted. According to the guidelines (<https://www.nature.com/commseng/submit/submission-guidelines#format-manuscripts>), specifically the word ,Introduction‘ should be avoided. However, after reviewing Communications Engineering articles, the title ‘Main‘ was substituted by ‘Introduction‘. Further editing of the content of both the introduction and the conclusions as well as the subtitles should assist in the understanding of the article.

The Conclusions section makes some statements regarding the importance of “this data” to the research community – but which data exactly is important and why is not given? What information is needed to de-risk the technology? What specific techno-economic barriers do industrial users need to be overcome have before adopting TES? That is, claims are made to the importance of this work and data presented but not really supported. For example, what does the readership now know that it critically did not know before reading this work as it pertains to TES and large-scale demonstrations?

Thank you for this insight. The conclusions were rewritten to give a better structure and better explain the concerns stated here.

Detailed Comments:

- 1- The Title is not meaningful. What about “superheated steam discharged” from MW TES? It reads like an unfinished phrase or point yet to be made. What about “Demonstration of superheated steam generation from MW-scale latent TES using molten salts”? or some such variation.

The title has been changed. As an explanation - in the original title, the authors were avoiding the term ‘Demonstrated‘ due to funding logistics based on the use of specific words (pilot, demonstrator, etc.), but understand that the verb “discharged” is not clear as the emphasis in the title.

- 2- Abstract – the authors refer to “power” levels – electrical or thermal? Be specific.
Clarification introduced, and made consistent throughout the article.

- 3- Abstract – the last sentences needs “, respectively” to be added to it.
Thank you, inserted.

- 4- Main, p. 2 - 3rd paragraph last sentence – what is the point of the last sentence? It is not clear why the readers should want to know that Garcia et al, have an extensive listing and discussion of latent heat storage at this particular location within your manuscript. Rather, a point should be made that is supported by Garcia, et al., and the reader can examine that reference more carefully is they want details on x, y or z.

This was rewritten and now reads:

In Garcia et al [4], an extensive listing and discussion of the existing high temperature latent heat storages is well documented; this listing shows developmental steps in the technology but no integration outside of the laboratory environment.

- 5- Main - How is thermal energy rate (“power”) defined? Provide definition.
The equation used for calculation is given.

- 6- Main, p. 2 – there is no reference for the two minute specification on GT lag time. Please provide.
A clarification regarding the two minutes is provided. It is, however, based on experience from the cogeneration plant operation. During this time after the GT trips, steam of sufficient quality is emitted from the HRSG, allowing for a startup of an auxiliary boiler. This requires that the power plant operator react very quickly. It now reads:
The steam must be provided by the standby system within the time that the HRSG produces steam after a GT-trip in lag-time, which is two minutes, as determined by experience from the power plant operator and discussed in [6]. This ensures a constant steam quality in the steam mains.
- 7- Text around Fig. 2 - The fin-tube assembly geometry is very intriguing. Recommend including 1-2 sentences on how this was developed (ie the geometry was established by what kind of objective function and conducting heat in/out of TES is a critically important challenge).
The authors appreciate your interest! Some explanation regarding the fin design is provided and it now reads as follows:
The fin design results from a combination of two-dimensional simulations of heat transfer, empirical experience in manufacturing extruded aluminum and experience with previous fin designs; the tube spacing in this storage is very dense at 70 mm.
- 8- p. 3 – sodium nitrate for the heat transfer fluid is given, but its thermophysical properties, namely is enthalpy change upon solidification and melting are not provided. Was this not determined from DSC? The supplemental information section provides no information on such thermodynamic property data obtained.
A clearer reference to materials research on sodium nitrate was included in the section regarding system design and integration, as well as to the DSC measurements in the supplementary information. The section now reads:
Due to its melting temperature between the system limitations of 300 °C and 350 °C, as well as proven thermal stability as a phase change material (PCM) [11], sodium nitrate is used. The relevant material properties, as discussed by [11], are a theoretical melting temperature of 306 °C, a specific heat capacity of 1655 J/(kg K), a heat conductivity of 0.55 W/(m K) and a latent heat of 178 kJ/kg. The melting temperature of the actual salt inventory was measured using differential scanning calorimetry as about 304 °C (supplementary information, Fig. 9).
- 9- p.3 – Why was sodium nitrate chosen as the HTF?
Explanation included for the use of sodium nitrate as the storage material; water/steam is the HTF. See response to comment #8 please.
- 10- Power capacity – “Thermal” power is a more accurate term. 2nd paragraph, last sentence – believe there is a type – “mass low” should be “mass flow”
Thank you, mass flow corrected. Thermal power consistently adjusted throughout the article.

REVIEWERS' COMMENTS:

Reviewer #1 (Remarks to the Author):

The paper is revised based on the comments and it is recommended for publication at this stage.

Reviewer #2 (Remarks to the Author):

The authors have addressed the comments, although the impact of the work has not been articulated as clearly as the reviewer recommended. It is publishable, although its suitability for Nature Communications is still a question.